# Antioxidant Active Polysaccharides Extracted with Oxalate from Wild Blackthorn Fruits (*Prunus spinosa* L.)

**DOI:** 10.3390/ijms25084519

**Published:** 2024-04-20

**Authors:** Peter Capek, Iveta Uhliariková

**Affiliations:** Institute of Chemistry, Slovak Academy of Sciences, Dúbravská cesta 9, SK-845 38 Bratislava, Slovakia

**Keywords:** wild blackthorn, *Prunus spinosa*, fruits, pectic polysaccharides, NMR

## Abstract

Although several therapeutic effects have been attributed to wild blackthorn fruits, their use is still negligible. Purification of the antioxidant-active fraction, obtained from wild blackthorn fruits by hot ammonium oxalate extraction (Ao), yielded seven fractions after successive elution with water, sodium chloride and sodium hydroxide solutions. The purified fractions differ in carbohydrates, proteins, and phenolics. About 60% of the applied Ao material was recovered from the column, with the highest yields eluted with 0.25 M NaCl solution, accounting for up to 70 wt% of all eluted material. Analyses have shown that two dominant fractions (3Fa and 3Fb) contain 72.8–81.1 wt% of galacturonic acids, indicating the prevalence of homogalacturonans (HG) with a low acetyl content and a high degree of esterification. The low content of rhamnose, arabinose and galactose residues in both fractions indicates the presence of RG-I associated with arabinogalactan. In terms of yield, the alkali-eluted fraction was also significant, as a dark brown-coloured material with a yield of ~15 wt% with the highest content of phenolic compounds of all fractions. However, it differs from other fractions in its powdery nature, which indicates a high content of salts that could not be removed by dialysis.

## 1. Introduction

In the Central European region (Slovakia), several wild fruits such as blackthorns (*Prunus spinosa* L.), rosehips (*Rosa canina*), hawthorns (*Crataegus monogyna*), rowans (*Sorbus aucuparia* L.), blueberries (*Vaccinium myrtillus* L.), cranberries (*Vaccinium vitis-idaea* L.), raspberries (*Rubus idaeus* L.), and blackberries (*Rubus fruticosus* L.) develop and ripen. Some of them are found abundantly, others less so. There is at present no social interest in the cultivation of these wild-growing shrubs, semi-shrubs and trees, or the possible use of their small and numerous fruits. They grow on uncultivated and neglected fields, meadows and pastures, in and near forests. Their characteristic features are the abundance of small edible fruits and their diverse colours. It has been found that these fruits are a source of a wide variety of biologically active compounds that have positive health effects on our bodies, such as antioxidants, anti-inflammatory agents, antimicrobials, etc. These edible berries are rich in antioxidants such as polyphenols, carotenoids, vitamins C and E, dietetic fibres, and others [1,2,3,4,5,6,7]. It is evident that this diverse range of wild fruits is not used enough, even though it shows a relatively large nutritional and functional potential. The reasons for the low interest in these fruits are their availability in natural locations and the difficulty of their collection and processing, which is related to the low area of their global use and means they are used only locally. In addition to phenolic compounds, which are the subject of considerable interest in the scientific community, carbohydrate polymers or their conjugates with protein and phenolic components are an important component of these edible berries, but little attention has been paid to them so far.

Recently, a preliminary study was carried out on wild blackthorn fruits from the point of view of the content of phenolic polysaccharide–protein complexes, their composition and antioxidant effects. Using different extraction agents, a series of phenolic polysaccharide–protein complexes differing in composition and antioxidant activity were isolated [8]. In addition, fractions extracted with cold and hot water were purified and characterized in more detail in terms of the types of polysaccharides present in them [9,10]. It should be remembered that, apart from phenolic compounds [11], relatively little attention has been paid to polysaccharides in blackthorn fruits.

As part of our research into the isolation and characterization of extractable phenolic polysaccharide–protein complexes from a little-explored area, such as the fruits of the wild blackthorn (*Prunus spinosa* L.), the current work is focused on the purification of the hot ammonium oxalate complex by ion exchange chromatography, the chemical characterization of purified fractions, and identification of their dominant polysaccharide components. In addition, the aim was to determine whether the native sample contains, in addition to RG-I, the presence of homogalacturonan sequences.

## 2. Results and Discussion

### 2.1. Ion-Exchange Chromatography and Chemical Characterization of Oxalate Fractions

In our previous work [8], we described the sequential extractions of crude polysaccharide complexes from ripe wild fruits of blackthorn (*Prunus spinosa* L.) with cold (Cw) and hot (Hw) water, hot 0.5% ammonium oxalate (Ao), 1% ammonia (Am), and finally 5% and 10% KOH solutions. Crude polysaccharides Cw and Hw were subsequently purified and characterized in more detail in terms of their polysaccharide composition [9,10]. Hot ammonium oxalate extraction (Ao) of wild blackthorn fruits yielded polysaccharide material in a yield of 1.7 wt% based on dried starting material, which was the third highest yield after Hw and 5% KOH. The polysaccharide complex Ao showed a much lower molecular weight (*M_w_* 54,900 g/mol) than Cw (*M_w_* 235,200 g/mol) or Hw (*M_w_* 218,400 g/mol). It was composed of carbohydrates (32.6 wt%), proteins (10.6 wt%), phenolics (8.5 wt%), and uronic acids (43.0 wt%). The highest content of uronic acid among all isolates was found in Ao. Of the neutral monosaccharides, which represented about 33 wt%, Ara (9.9 wt%) and Gal (7.6%) had the highest content, while other sugars such as Glc, Man, Rha, Xyl, and Fuc were found only in low abundance. In addition, Ao was found to exhibit significant antioxidant activity (Figure 1), much higher than Cw and comparable to Hw [8].

In order to characterise the oxalate complex in more detail, it was subjected to ion exchange chromatography (Figure 2). Elution with water (1F), NaCl solutions of increasing ionic strength (2F–5F), and finally with 1M sodium hydroxide solution (6F) provided fractions that differed in the content of carbohydrates, proteins, phenolics, uronic acids (Table 1), as well as neutral monosaccharides (Table 2). It was found that up to 40% of Ao was irreversibly bound and could not be eluted even with 1.0 M NaOH solution. This dark brown material bound at the top of the column comes from phenolics and highly charged acidic polysaccharides. After the purification process, fractions 1F–4F were white and almost free of phenolics. Sample 5F was light brown and only 6F was dark brown, indicating a high phenolic content. Regarding yields, the fractions eluted with 0.25 M NaCl (3Fa and 3Fb) and 1M NaOH solutions made up 51 wt% of all eluted material, and only about 9 wt% represented the other fractions (Table 1).

The first non-retained fraction (1F), eluted from the column with water, represented neutral polysaccharides such as arabinogalactan, and polysaccharides rich in Glc and Man residues. However, the yield (2%) of this fraction was negligible (Table 2). The following fraction (2F), eluted from the column by 0.1 M NaCl, differed from the previous fraction mainly in the content of uronic acids (41.9 wt%, Table 1), which indicates a high content of pectic polymers. The low Rha (1.2 wt%, Table 2) and high uronic acid (41.9 wt%) contents indicate the dominance of homogalacturonide (HG) in this fraction. In addition, arabinogalactan (~12 wt%) and polysaccharides containing Glc and Man residues (~16 wt%) were also present. Similar to 1F, the yield of this fraction (3.4 wt%) was also low (Table 1). Elution with 0.25 M NaCl yielded two dominant fractions, 3Fa and 3Fb, which represented up to 70% of all eluted material. Both fractions did not contain or had a negligible content of proteins and phenolics, and showed the highest content of GalA (72.8–81.1 wt%, Table 1) of all fractions, which indicates the dominance of homogalacturonans (HG) in these purified fractions. The low contents of Rha in both fractions indicate the presence of RG-I polymers as well, but only in small amounts, similar to the content of arabinogalactans (6–9%), which are usually associated with RG-I as side chains. Materials eluted with 0.5 and 1.0 M NaCl solutions (4F and 5F) were also rich in uronic acids (60 and 40.4 wt%, Table 1) and also indicated the dominance of HG, similar to the previous fractions. In addition, the light brown material (5F) contained a low content of RG-I (~12 wt%) and also arabinogalactan (~13 wt%). However, in terms of yield, both fractions were negligible. The last dark brown material, eluted from the column with 1.0 M NaOH (6F) had the third highest yield (~9 wt%). This dark-coloured material had a high phenolic content, much higher than the unfractionated Ao sample, and differed from the other fractions in its powdery character, indicating a high content of salts that could not be removed by dialysis. In addition, it also contained some proteins, and a low content of uronic acids and neutral sugars, which is indicated by a low arabinogalactan and RG-I components.

### 2.2. NMR Spectroscopy of Ammonium Oxalate Fractions (Ao) from Fruits of P. spinose

^1^H NMR spectra of Ao and its ion exchange fractions (1F–6F) are shown in Figure 3. All spectra were measured at the same conditions, and the internal standard TSP was used for the chemical shift calibrations. Signals of carbohydrates at ~δ_H_ 5.4–3.3 ppm were dominant, while those of proteins at ~δ_H_ 2.6–1.0 ppm were of low intensity. Phenolic signals at ~δ_H_ 8.0–6.0 ppm were recorded as very broad signals.The presence of deoxy-sugars i.e., Rha and Fuc as doublets in the range of δ_H_ 1.2–1.4 ppm, were visible in all fractions and increased in intensity from the aqueous fraction (1F) to the hydroxide fraction (6F). In addition, signals of acetyl groups in the range of δ_H_ 2.2–2.0 ppm (2F–5F) and the intense methyl group signal at δ_H_ 3.8 ppm (2F–5F), common for esterified carboxyl groups of GalA in pectin compounds, were observed. From Figure 3, it is evident that the α-type of glycosidic bond is predominant in all samples.

HSQC spectra of Ao and its ion exchange fractions (Figure 4) provide us with more information about their structural characteristics, such as the variability of sugar bonds, their content, anomeric character, and the presence and intensity of substituents. In comparison with the literature data of pectic polysaccharides [12,13,14,15,16,17,18], the most intense broad signals indicated that the polysaccharide backbones of Ao and its ion exchange fractions contain regions of methoxyl and acetyl 1,4-α-D-galacturonan and rhamnogalacturonan. The side chains of the branched region consist of differently linked α-Ara*f* and β-Gal*p* residues, suggesting the presence of highly branched arabinan or arabinogalactan. The dominant fraction 3Fa with the highest content of GalA was used for detailed NMR characterization. 2D ^1^H-^1^H homo-correlated (COSY, TOCSY) and ^1^H-^13^C hetero-correlated (HSQC, HSQC-TOCSY, HMBC) experiments were used for the identification of individual sugar residues spin systems. The obtained NMR data are summarized in Table 3.

In the HSQC spectrum of 3Fa fraction (Figure 5), the dominant broad anomeric H1/C1 cross peaks at δ_H1/C1_ 4.929/102.97, 4.956/103.07, 5.119/102.24, 5.127/101.75 ppm originate from α-GalA*p* residues and its methyl ester located in different environments of the HG. The intense signal of a methyl group at δ_H/C_ 3.809/55.75 ppm confirmed the methyl esterification of α-GalA*p* residues [14,15,16,17]. Further, characteristic H5/C5 signals of esterified α-GalA*p* (GalA^a,b^, Figure 5, Table 3; δ_H5/C5_ 5.055/73.48, 5.108/73.42 ppm) and non-esterified residues (GalA^c,d^, Figure 5, Table 3; δ_H5/C5_ 4.687/74.32, 4.725/74.48 ppm) were observed. The ratio of esterified to non-esterified residues 62:38 indicated a high degree of esterification. In the HMBC spectrum, the observed long-range correlations between α-GalA*p* anomeric protons and their C4 carbons (GalA^a,b,c,d^, Table 3; δ_H1/C4_ 4.929/81.32, 4.956/81.68, 5.119/81.58, 5.127/80.69 ppm) confirmed the 1,4-linkage of α-GalA*p* residues of the HG moieties. According to the literature data, two cross peaks of acetyl groups observed at δ_H/C_ 2.086/23.14 and 2.176/23.40 ppm suggested the presence of acetylated α-GalA*p* residues at position O2 and O3, respectively, in this pectin material [18,19]. Moreover, signals due to the reducing end of GalA*p* residues, α-GalA*p* (GalA^e^, Figure 5, Table 3; δ_H1/C1_ 5.318/95.11 ppm) and β-GalA*p* (GalA^f^, Figure 5, Table 3; δ_H1/C1_ 4.957/99.13 ppm) were identified.

Further, the anomeric cross peak at δ_H1/C1_ 5.279/101.36 ppm was attributed to α-Rha*p* residue in rhamnogalacturonan (RG-I). Its presence confirmed also a characteristic H6/C6 signal at δ_H6/C6_ 1.249/19.51 ppm (Figure 5, Table 3). The 1,2-linkage of α-Rha*p* was identified by its downfield-shifted C2 atom chemical shift at δ_C2_ 79.29 ppm [14,15,16,17].

In addition, the HSQC spectrum showed six less intense anomeric H1/C1 cross peaks of α-Ara*f* residues in the region δ_H1/C1_ 5.079–5.247/109.38–112.09 ppm [9,12,13]. The anomeric signal at δ_H1/C1_ 5.079/110.48 ppm (Ara^a^, Figure 5, Table 3) was assigned to 1,5-linked α-Ara*f*. The linkage was confirmed by its downfield-shifted C5 atom (δ_C5_ 69.81 ppm) with respect to the terminally linked Ara*f* residues (Table 3) and also by the HMBC correlation between its H1 and C5 atoms (δ_H1/C5_ 5.079/69.81 ppm). The anomeric signal at δ_H1/C1_ 5.105/110.49 ppm was assigned to 1,3,5-linked α-Ara*f* residue. Its downfield-shifted C3 atom (3Ara^b^, δ_C3_ 85.38 ppm, Figure 5, Table 3) suggested the substitution of this sugar residue at position O3. The chemical shift of the C5 atom (5Ara^b^, δ_C5_ 69.37 ppm, Figure 5, Table 3) and the observed HMBC correlation δ_H1/C5_ 5.105/69.37 ppm confirmed the 1,5-linkage of Ara^b^. The HSQC spectrum also revealed the presence of 1,2,3,5-linked α-Ara*f* residue (Ara^c^, δ_H1/C1_ 5.238/109.38 ppm, Figure 5, Table 3) with the characteristic downfield chemical shifts of its C2, C3 and C5 atoms (δ_C2_ 87.99 ppm, δ_C3_ 83.39 ppm, δ_C5_ 69.54 ppm). Further, three anomeric signals of terminally linked α-Ara*f* residues were found (Ara^T^, Figure 5, Table 3). These Ara^T^ can be linked either to different sugar residues or to identical residues but in different positions. Due to overlapping signals in the HMBC spectrum, only the long-range correlation of the dominant one Ara^T^ residue (δ_H1/C1_ 5.148/110.06, Figure 5, Table 3) from its H1 proton to O3 position of Ara^b^ (δ_H1/C_ 5.148/85.38 ppm) was identified, thus confirming the 1,3,5-linkage of Ara^b^ residue. Further, the anomeric cross peak at δ_H1/C1_ 4.619/107.26 ppm was assigned to 1,4-linked β-Gal*p* residue, which was proved by the HMBC correlation of its H1 and C4 atoms at δ_H1/C4_ 4.619/80.49 ppm. β-Gal*p* together with α-Ara*f* residues indicated a high binding variability of these saccharides in the arabinan or arabinogalactan moieties.

## 3. Materials and Methods

### 3.1. Plant Material and Isolation of Phenolic Polysaccharide–Protein Complexes from Blackthorn Fruits

Harvested ripe fruits (collected in the Kysuce region, Svrčinovec-Bahno district, Slovakia) of wild *Prunus spinosa* L. were pitted, mixed, and juiced (3.6 kg). The residues were boiled in 96% ethanol (3 L, 30 min), washed, and freeze-dried after mixing in distilled water to give crude starting material of fruits (510 g) for extraction processes [8].

The freeze-dried dark blue-coloured blackthorn residues (500 g) were stirred three times in distilled water (3 L) at laboratory temperature for 20 h in the presence of a 0.02% solution of sodium azide. The extracts were filtered, combined, concentrated to lower volumes and precipitated in 96% ethanol (1: 5 *v*/*v*) acidified with 1% of acetic acid. The precipitate was removed by centrifugation, washed with ethanol, dissolved in distilled water, dialyzed and lyophilized to obtain the cold water fraction (4.8 g), labelled as Cw. 

The residue after cold water extraction was further stirred three times in hot distilled water (70 °C, 3 L) for 1 h. The extracts were filtered, combined, concentrated, and precipitated in 96% ethanol (1: 5 *v*/*v*) acidified with 1% of acetic acid. The precipitate was washed with ethanol, dissolved in distilled water, dialyzed, and lyophilized to obtain a hot water fraction (19.1 g), labelled as Hw.

The remaining residue after cold and hot water extractions was further extracted three times in 0.5% aqueous ammonium oxalate (2.5 L) for 2 h at 90 °C. The extracts were filtered, combined, neutralized, concentrated to lower volumes and precipitated in 96% ethanol (1: 5 *v*/*v*) containing 1% of acetic acid. The precipitate was removed by centrifugation, washed with ethanol, dissolved in distilled water, dialyzed, and freeze-dried to give an ammonium oxalate fraction (8.3 g), labelled as Ao.

### 3.2. Isolation and Purification of the Hot Ammonium Extract (Ao) from Blackthorn Berries

The crude polysaccharide complex extracted with ammonium oxalate (Ao, 2 g) was dissolved in distilled water (300 mL), the insoluble part was removed by centrifugation and the soluble sample was loaded on a DEAE Sephacel column (4.5 × 16 cm). The column was eluted successively with distilled water (1F), 0.1 (2F), 0.25 (3Fa, 3Fb), 0.5 (4F), and 1.0 M (5F) NaCl solutions, and finally 1.0 M NaOH solution (6F, containing 10 mM NaBH_4_ solution). The elution rate was 15 mL/h and fractions of 5 mL were collected (excluding the sodium hydroxide fraction collected in the bakery) and analysed for carbohydrate content by the phenol–sulphuric acid test [20] to obtain the elution carbohydrate profile (Figure 1). The tubes of the individual fractions were pooled, dialysed (MWCO 1000, Spectra/Por, Los Angeles, CA, USA) in tap and distilled water for three days, and freeze-dried.

### 3.3. General Methods

The content of carbohydrates, proteins, uronic acids and phenols was determined colourimetrically using phenolic sulphuric acid, Bradford, m-hydroxydiphenyl, and Folin–Ciocalteu tests [20,21,22,23]. Glucose, serum albumin, galacturonic acid, and gallic acid, respectively, were used as standards. Colourimetric tests were measured with a UV-VIS 1800 spectrophotometer (Shimadzu, Kyoto, Japan).

The quantitative composition of neutral sugars in the samples was determined after total hydrolysis with 2M trifluoroacetic acid at 120 °C for 1 h, reduction of NaBH_4_, and acetylation by gas chromatography as alditol acetates [24] on a TRACE Ultra Gas Chromatograph (Thermo Scientific, Waltham, MA, USA) equipped with a TG-SQC capillary column (Thermo Scientific, Waltham, MA, USA) (30 m × 0.25 mm × 0.2 μm) at a temperature program of 80 °C (4 min)–(8 °C/min)–160 °C (4 min)–(4 °C/min)–250 °C (20 min). The helium flow rate was 0.4 mL/min. The gas chromatograph was connected to an ITQ 900 mass spectrometer (Thermo Scientific, Waltham, MA, USA) with EI ionization at a standard electron energy of 70 eV, an emission current of 25 μA, and an ion source temperature of 200 °C. A mixture of rhamnose, fucose, ribose, arabinose, xylose, mannose, galactose, and glucose alditol acetates was used as a standard.

NMR spectra (^1^H and ^1^H-^13^C HSQC) were measured in D_2_O at 60 °C on a Bruker AVANCE III HDX 400 MHz spectrometer (Bruker, Reutlinger, Germany) equipped with a broad band BB-(H–F)-D-05-Z liquid N_2_ Prodigy probe (Bruker BioSpin, Karlsruhe, Germany), using 3-methyl-silyl-propionic acid sodium salt (Armar Chemicals, Dottingen, Switzerland)—TSP-d_4_ (δ H/C 0,0 ppm) as an internal standard. Each sample was frieze-dried twice (99.90% D_2_O) before measurement. The detailed NMR analysis of fraction 3Fa was performed at 60 °C on Bruker AVANCE III HDX 600 MHz spectrometer (Bruker, Reutlinger, Germany) equipped with a Triple resonance TCI H-C/N-D-05-Z, liquid He cooled cryo probe (Bruker BioSpin, Karlsruhe, Germany). For signals assignment advanced techniques of 1D and 2D homo- and hetero-correlated spectroscopy from Bruker pulse sequence library were used.

### 3.4. Statistical Analyses

Statistical analyses of results were expressed as mean ± standard deviation. They were statistically analysed using Microsoft Office Excel LTSC Standard 2021 software (Microsoft Corporation, Redmond, WA, USA). Dixon’s Q test was used for the identification and rejection of outliers. Differences were considered statistically significant when *p* < 0.05 (*n* = 3).

## 4. Conclusions

The crude polysaccharide complex, isolated from ripe berries of wild *Prunus spinosa* by hot extraction of ammonium oxalate, was shown after separation by ion exchange chromatography to be mainly a pectin material. Some dominant ion exchange fractions represent typical pectin polymers with a high content of homogalacturonan and a minor content of rhamnogalacturonan components. In general, pectin is widely used in the food industry as a gelling, thickening, and emulsifying agent. In addition to the food industry, it is also used in the medical and pharmaceutical industries. From this point of view, these underutilized wild blackthorn fruits could be a potential source of pectin materials. Wild blackthorn fruits have a high content of pectic polysaccharides and phenolic compounds, which increases their attractiveness as nutritional and especially antioxidant substances. The antioxidant effect of phenolic polysaccharide complexes from wild blackthorn fruits may promote interest in these undiscovered natural fruits as a source of natural antioxidants.

## Figures and Tables

**Figure 1 ijms-25-04519-f001:**
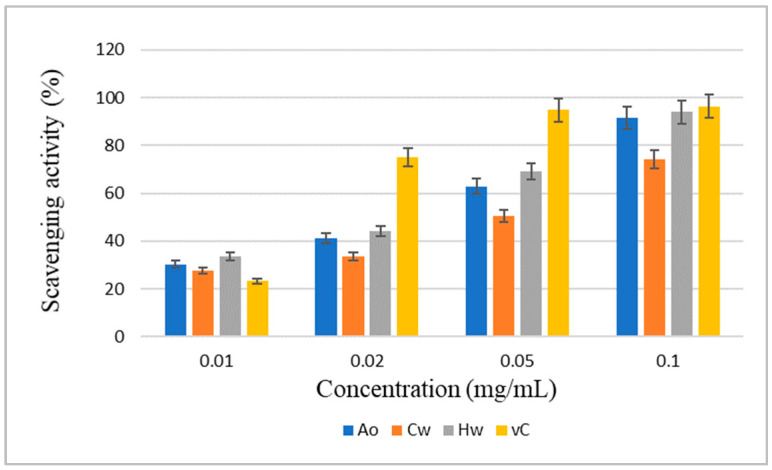
Comparison of antioxidant activity of *P. spinosa* fraction Ao (hot ammonium oxalate extract) with Cw (cold water extract), Hw (hot water extract), and vitamin C (vC).

**Figure 2 ijms-25-04519-f002:**
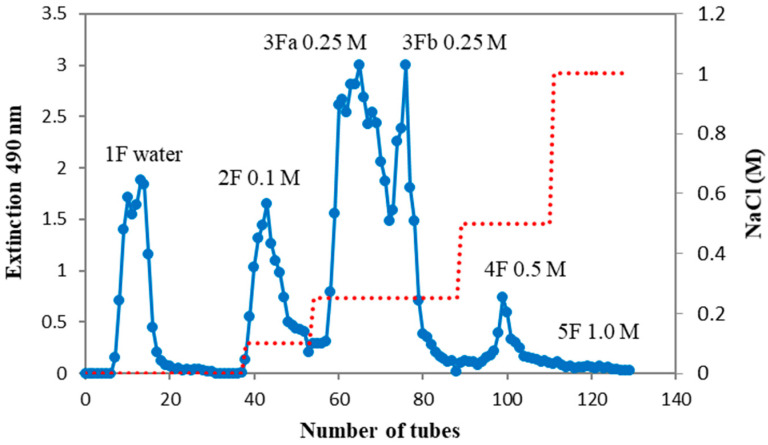
Ion exchange chromatography of a hot ammonium oxalate extract (Ao) of wild *P. spinosa* fruits.

**Figure 3 ijms-25-04519-f003:**
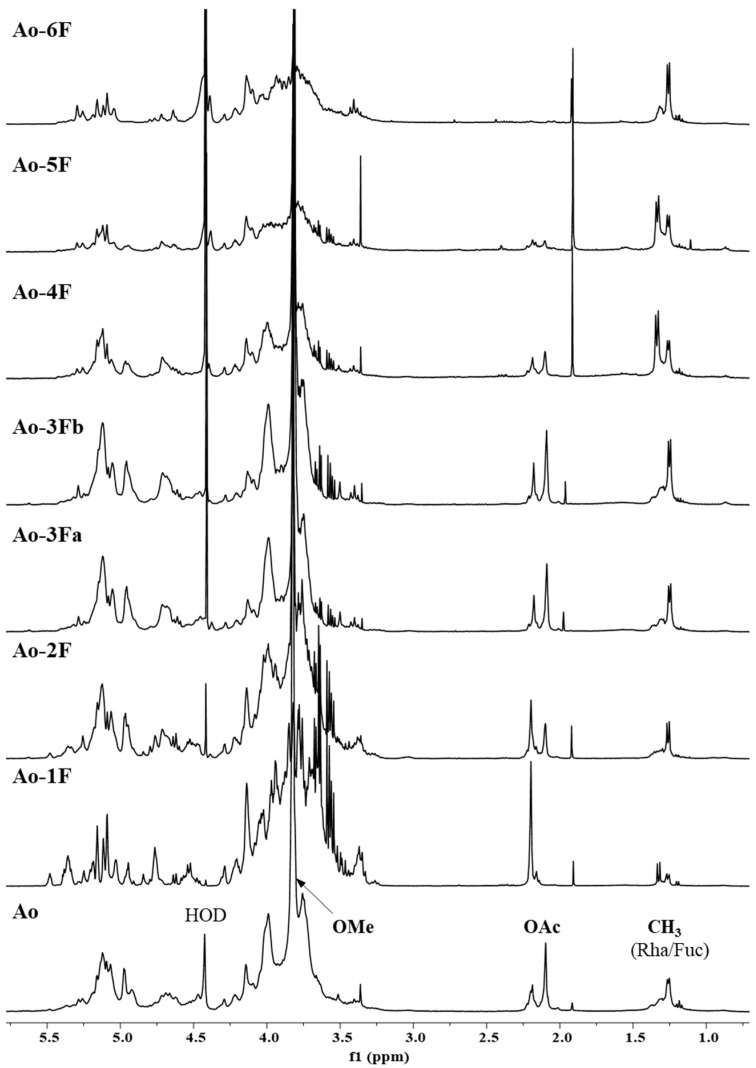
The selected region of ^1^H NMR spectra of the ion exchange fractions (1F–6F) of hot ammonium oxalate extract (Ao) from the wild fruits of *P. spinose*. HOD—residual signal of water; OAc—*O*-acetyl; OMe—*O*-methyl.

**Figure 4 ijms-25-04519-f004:**
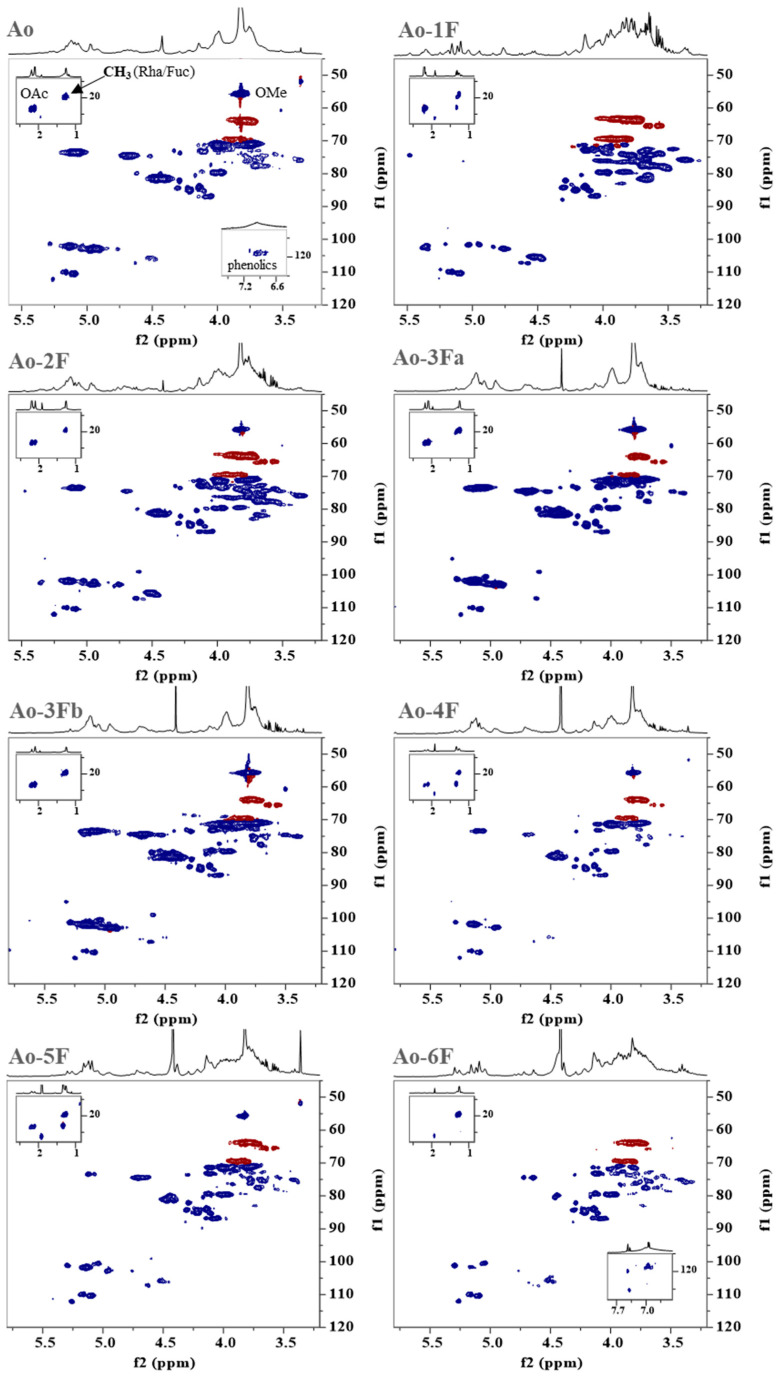
Selected regions of ^1^H-^13^C hetero-correlated HSQC spectra of the ion exchange fractions (1F–6F) of the hot ammonium oxalate extract (Ao) from the wild fruits of *P. spinose*. OAc—*O*-acetyl; OMe—*O*-methyl; CH—blue colour; CH_2_—red colour.

**Figure 5 ijms-25-04519-f005:**
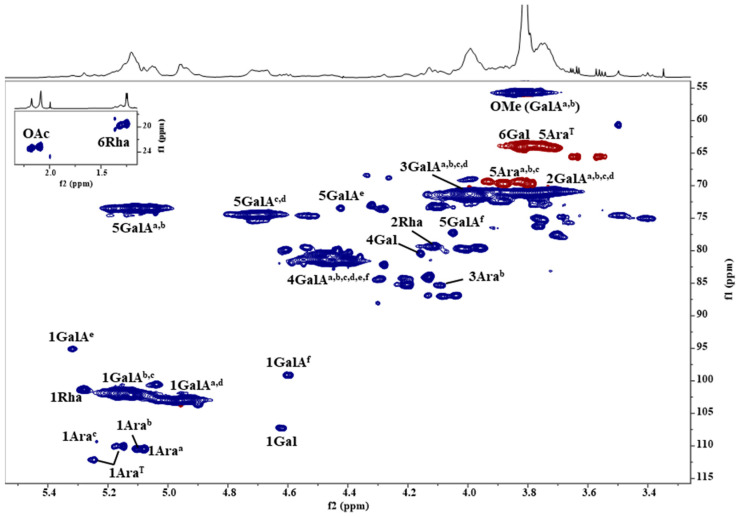
The selected region of HSQC spectrum of 3Fa fraction of the hot ammonium oxalate extract (Ao) from the wild fruits of *P. spinose*. The saccharide labels and their linkages are explained in Table 3. OAc—*O*-acetyl; OMe—*O*-methyl; CH—blue colour; CH_2_—red colour.

**Table 1 ijms-25-04519-t001:** Compositional analyses of ion exchange fractions of ammonium oxalate (Ao) from wild *P. spinosa* L. fruits.

Fraction	Yield	Carb ^c^	Pro ^d^	Phe ^e^	UA ^f^
	mg/wt%	(wt%)			
Native Ao	2000/100	32.6 ± 0.38	10.6 ± 0.12	8.5 ± 0.07	43.0 ± 0.06
1F water	40/2.0	50.3 ± 0.26	1.4 ± 0.04	0.0 ± 0.0	1.0 ± 0.08
2F 0.1 M ^a^	68/3.4	28.2 ± 0.42	0.4 ± 0,05	0.0 ± 0.0	41.9 ± 0.0
3Fa 0.25 M ^a^	565/28.3	19.7 ± 0.18	0.1 ± 0.06	0.0 ± 0.0	81.1 ± 0.32
3Fb 0.25 M ^a^	280/14.0	14.3 ± 0.53	0.1 ± 0.11	0.2 ± 0.04	72.8 ± 0.11
4F 0.5 M ^a^	52/2.6	15.0 ± 0.46	4.4 ± 0.12	0.3 ± 0.07	60.1 ± 0.87
5F 1.0 M ^a^	7.0/0.4	19.6 ± 0.22	1.8 ± 0.03	2.4 ± 0.08	40.4 ± 0.18
6F 1.0 M ^b^	175/8.8	14.0 ± 0.14	4.0 ± 0.08	15.9 ± 0.23	4.9 ± 0.04

^a^ 0.1–1.0 M NaCl, ^b^ 1.0 M NaOH, ^c^ Carbohydrate, ^d^ Protein, ^e^ Phenolic and ^f^ Uronic acid contents, wt%: weight percent.

**Table 2 ijms-25-04519-t002:** Monosaccharide compositional analyses of ion exchange fractions of ammonium oxalate (Ao) from wild *P. spinosa* L. fruits.

Fraction	Neutral Monosaccharide Composition ^c^ (wt%)						
	L-Rha	L-Fuc	L-Ara	D-Xyl	D-Man	D-Gal	D-Glc
Native Ao	3.5	1.1	9.9	1.7	3.9	7.6	4.9
1F water	0.3	0.2	9.0	1.1	11.5	7.8	22.9
2F 0.1 M ^a^	1.2	0.1	4.9	1.2	7.6	6.7	8.1
3Fa 0.25 M ^a^	4.1	0.3	5.5	1.3	1.8	3.6	2.0
3Fb 0.25 M ^a^	5.4	0.4	3.8	0.9	1.0	2.5	0.9
4F 0.5 M ^a^	2.4	0.2	3.5	0.7	2.5	2.6	4.4
5F 1.0 M ^a^	5.8	0.3	8.3	1.6	2.4	5.0	2.4
6F 1.0 M ^b^	1.6	tr.	6.8	0.8	1.2	3.4	1.9

^a^ 0.1–1.0 M NaCl, ^b^ 1.0 M NaOH, ^c^ Converted to the content of carbohydrates (32.6 wt%, Table 1) determined by the phenol-sulfuric method, wt%: weight percent. The results of the monosaccharide compositions of the fractions are expressed as the average of two measurements.

**Table 3 ijms-25-04519-t003:** ^1^H and ^13^C NMR data of 3Fa fraction of the hot ammonium oxalate extract (Ao) from the wild fruits of *P. spinose*.

Residue	Label	Chemical Shift δ (ppm)						
		H1/C1	H2/C2	H3/C3	H4/C4	H5, H5′/C5	H6, H6′/C6	OMe
→4)-α-GalA*p*-6-OMe(1→	GalA ^a^	4.956	3.731	3.982	4.453	5.055	-	3.809
		103.07	70.88	70.90	81.69	73.48	173.59	55.75
→4)-α-GalA*p*-6-OMe(1→	GalA ^b^	5.119	3.732	4.004	4.404	5.108	-	3.809
		102.24	71.33	71.05	81.58	73.42	173.66	55.75
→4)-α-GalA*p*(1→	GalA ^c^	5.127	3.751	4.003	4.450	4.725	-	
		101.75	71.12	71.69	80.69	74.48	177.69	
→4)-α-GalA*p*(1→	GalA ^d^	4.929	3.743	3.965	4.478	4.687	-	
		102.97	70.87	71.61	81.32	74.32	177.60	
→4)-α-GalA*p*	GalA ^e^	5.318	3.818	3.993	4.432	4.423	-	
		95.11	71.07	71.62	81.08	73.51	177.73	
→4)-β-GalA*p*	GalA ^f^	4.597	3.490	3.752	4.359	4.050	-	
		99.13	74.53	75.33	80.86	77.23	177.03	
→2)-α-Rha*p*(1→	Rha	5.279	4.112	3.882	3.401	3.766	1.249	
		101.36	79.29	72.14	75.02	75.03	19.51	
→4)-β-Gal*p*(1→	Gal	4.619	3.682	3.761	4.156	3.684	3.820, 3.791	
		107.26	74.82	76.28	80.4	77.98	63.90	
→5)-α-Ara*f*(1→	Ara ^a^	5.079	4.128	4.002	4.202	3.881, 3.792		
		110.48	83.91	79.79	85.20	69.81		
→3,5)-α-Ara*f*(1→	Ara ^b^	5.105	4.282	4.092	4.297	3.931, 3.830		
		110.49	82.26	85.38	84.46	69.37		
→2,3,5)-α-Ara*f*(1→	Ara ^c^	5.238	4.290	4.230	n.d.	3.937, 3.840		
		109.38	87.99	83.39		69.54		
α-Ara*f*(1→	Ara ^T^	5.148	4.130	3.958	4.039	3.821, 3.724		
		110.06	84.22	79.61	86.88	64.19		
α-Ara*f*(1→	Ara ^T^	5.171	4.125	n.d.	n.d.	n.d.		
		109.99	84.31	79.57				
α-Ara*f*(1→	Ara ^T^	5.247	4.210	3.974	n.d.	n.d.		
		112.09	84.25	79.64				

OMe—*O*-methyl, n.d.—not detected. ^a-f^ GalA: the sugar units in different environments and bindings. ^a-c,T^ Ara: different linked sugar units.

## Data Availability

The data presented in this study are available upon request from the corresponding author.

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
