# Peer review of "Antioxidant Active Polysaccharides Extracted with Oxalate from Wild Blackthorn Fruits (Prunus spinosa L.)"

_ijms, 2024, doi:10.3390/ijms25084519_

Round 1

Reviewer 1 Report

Comments and Suggestions for Authors

Article Antioxidant active polysaccharides extracted with oxalate from wild blackthorn fruits (Prunus spinosa L.) by authors Peter Capek and Iveta Uhliariková examines the composition of Prunus spinosa metabolites in relation to their beneficial properties.

The work was carried out with a current and popular object, which is included in a number of food products, soybeans, and is considered as a valuable functional food and traditional medicine.

The authors have carried out work on extraction, purification and analysis, which are of well-known interest. But there are a number of problems; there is no statistical processing in the manuscript, no qualitative characteristics of the object and no conditional “control”. Nevertheless, the data may be interesting in short message mode when adding statistics and correcting a number of design violations.

Authors should expand the introduction by describing research options in this area and the potential benefits and limitations of methodological approaches.

In the last paragraph, place the hypothesis or task to which the work is devoted and not a link.

In Table 1, indicate the number of repetitions and the spread in the data for different measurements and other statistics, after all, didn’t the authors do this once?

The description of Figure 3 should describe what constituted a reliable detection control.

In Figure 4, increase the magnification and resolution, expand the legend so that the method, approach and result are clear.

In the methods, expand the description of sampling and sample preparation.

In conclusion, describe the main conclusion(s) and prospects, and move the results into discussion.

In Figures 4 and 5, some parts of the graph are cut off, please correct this.

The article can be accepted after editing.

Author Response

Manuscript ID: ijms-2918627

Title: Antioxidant active polysaccharides extracted with oxalate from wild blackthorn fruits (Prunus spinosa L.)

Authors: Peter Capek *, Iveta Uhliariková

Author's Reply to the Review Report (Reviewer 1)

Comments and Suggestions for Authors

Article Antioxidant active polysaccharides extracted with oxalate from wild blackthorn fruits (Prunus spinosa L.) by authors Peter Capek and Iveta Uhliariková examines the composition of Prunus spinosa metabolites in relation to their beneficial properties.

The work was carried out with a current and popular object, which is included in a number of food products, soybeans, and is considered as a valuable functional food and traditional medicine.

The authors have carried out work on extraction, purification and analysis, which are of well-known interest. But there are a number of problems; there is no statistical processing in the manuscript, no qualitative characteristics of the object and no conditional “control”. Nevertheless, the data may be interesting in short message mode when adding statistics and correcting a number of design violations.

Authors should expand the introduction by describing research options in this area and the potential benefits and limitations of methodological approaches.

Answer:

The Introduction section has been expanded.

In the last paragraph, place the hypothesis or task to which the work is devoted and not a link.

Answer:

According to the reviewer's comment, the literary reference was removed and the task was added.

In Table 1, indicate the number of repetitions and the spread in the data for different measurements and other statistics, after all, didn’t the authors do this once?

Answer:

In Table 1, the data were calculated as the average of 3 measurements. Now we have calculated and added the statistical deviations. In Table 2 are results of monosaccharide analysis of fractions. In this case, the monosaccharide analysis was measured twice and the results are expressed as the average of two measurements without statistical deviations.

The description of Figure 3 should describe what constituted a reliable detection control.

Answer:

NMR spectroscopy is the highly reproducible and a high throughput technology, for which not detection control is usually needed in structural studies. Analyses were performed on 600MHz spectrometer equipped with cryo probe, which increases the sensitivity of the measurements. For all samples the chosen quantity taken for an analysis was sufficient and comparable. From this point of view no comment about detection control is needed in the description of the Figure 3. Comment of the Figure 3 was made according standard descriptions used for 1H NMR spectra of polysaccharides.

In Figure 4, increase the magnification and resolution, expand the legend so that the method, approach and result are clear.

Answer:

According to the reviewer's comment, Figure 4 has been enlarged and the legend has been improved.

In the methods, expand the description of sampling and sample preparation.

Answer:

Data were added in the Materials and Methods section.

In conclusion, describe the main conclusion(s) and prospects, and move the results into discussion.

Answer:

According to the reviewer's comment, conclusions were corrected.

In Figures 4 and 5, some parts of the graph are cut off, please correct this.

Answer:

We tried to display more details in the carbohydrate parts of the HSQC spectra for a better comparison of all fractions, therefore we did not deal with phenols. Sufficiently high content of phenolic substances with detectable signals in the HSQC spectra was only in the native sample Ao and in fraction 6F (after increasing the spectral intensity). Therefore, Figure 4 was modified by additional inserts of phenolic moieties in the HSQC spectra of these two samples.

Figure: The whole 1H-13C hetero-correlated HSQC spectrum of hot ammonium oxalate extract Ao from the wild fruits of P. spinose (A); (B) – increased spectral intensity.

The article can be accepted after editing.

Reviewer 2 Report

Comments and Suggestions for Authors

This paper described the isolation and characterization of extractable phenolic polysaccharide-protein complexes from a little-explored area, such as the fruits
of wild blackthorn (Prunus spinosa L.). This work is focused on purifying
the hot ammonium oxalate complex by ion exchange chromatography, the chemical characterization of purified fractions, and identifying their dominant polysaccharide components. The isolation method and characterization using 2D-NMR are useful and interesting to the readers. Thus, it is recommended that this paper be published in IJMS after minor revisions.

Minor
Please ensure their color's role in the HSQC spectrum can be added to the memo in the legend.

Author Response

Manuscript ID: ijms-2918627

Title: Antioxidant active polysaccharides extracted with oxalate from wild blackthorn fruits (Prunus spinosa L.)

Authors: Peter Capek *, Iveta Uhliariková

Author's Reply to the Review Report (Reviewer 2)

Comments and Suggestions for Authors
This paper described the isolation and characterization of extractable phenolic polysaccharide-protein complexes from a little-explored area, such as the fruits
of wild blackthorn (Prunus spinosa L.). This work is focused on purifying
the hot ammonium oxalate complex by ion exchange chromatography, the chemical characterization of purified fractions, and identifying their dominant polysaccharide components. The isolation method and characterization using 2D-NMR are useful and interesting to the readers. Thus, it is recommended that this paper be published in IJMS after minor revisions.

Please ensure their color's role in the HSQC spectrum can be added to the memo in the legend.
Answer:

The signal’s color description was added to the legend of Figures.

Reviewer 3 Report

Comments and Suggestions for Authors

The publication has no novelty. It is interesting but should not be published in a good journal like IJMS. The authors should try submitting it to another journal with a lower IF.

Justification for my decision to reject the publication:

The authors use standard methods and techniques, and have not developed any method themselves (there is no novelty here). The authors study standard material for which analogous studies have already been conducted and the results are published. Although the results of the study are interesting, their quality is too low to appear in IJMS.

Author Response

Manuscript ID: ijms-2918627

Title: Antioxidant active polysaccharides extracted with oxalate from wild blackthorn fruits (Prunus spinosa L.)

Authors: Peter Capek *, Iveta Uhliariková

Author's Reply to the Review Report (Reviewer 3)

Comments and Suggestions for Authors

The publication has no novelty. It is interesting but should not be published in a good journal like IJMS. The authors should try submitting it to another journal with a lower IF.

Justification for my decision to reject the publication:

The authors use standard methods and techniques, and have not developed any method themselves (there is no novelty here). The authors study standard material for which analogous studies have already been conducted and the results are published. Although the results of the study are interesting, their quality is too low to appear in IJMS.

Answer:

The results of ion-exchange chromatography of an ammonium oxalate sample, isolated from wild Prunus spinosa fruits have not yet been published.

Round 2

Reviewer 1 Report

Comments and Suggestions for Authors

Article Antioxidant active polysaccharides extracted with oxalate from wild blackthorn fruits (Prunus spinosa L.) by authors Peter Capek, Iveta Uhliarikova describe interesting results of analyzes of blackthorn metabolites.

The authors made corrections, reorganized the materials and methods section and the conclusion.

In the results section, the authors made corrections to the figures and captions.

The article has been improved and can be published in its present form.

Reviewer 3 Report

Comments and Suggestions for Authors

After the revision, I do not change my original decision. I suggest rejecting the publication. It has only elements of novelty. It should be published but not in IJMS, but in a journal with a lower IF. However, I understand that the final decision belongs to the editor and I will respect it.